# Zootechnical Performance and Some Physiological Indices of Tambaqui, *Colossoma macropomum* Juveniles during Biofloc Maturation and in Different Feed Regimes

Cintia Labussière Nakayama [1], Luiz Felipe Silveira Silva [1], Fabio Aremil Costa dos Santos [1], Tulio Pacheco Boaventura [1], Gisele Cristina Favero [1], Glauber David Almeida Palheta [2], Nuno Filipe Alves Correia de Melo [2], Luiz Alberto Romano [3] and Ronald Kennedy Luz [1,*]

[1] Laboratório de Aquacultura, Departamento de Zootecnia, Escola de Veterinária, Universidade Federal de Minas Gerais-UFMG, Avenida Antônio Carlos 6627, Belo Horizonte CEP 31270-901, MG, Brazil; cintianakayama@vetufmg.edu.br (C.L.N.); luizssilva2209@vetufmg.edu.br (L.F.S.S.); fabioaremil@vetufmg.edu.br (F.A.C.d.S.); tuliopb@ufmg.br (T.P.B.); giselefavero@vetufmg.edu.br (G.C.F.)

[2] Programa de Pós-Graduação em Aquicultura e Recursos Aquáticos Tropicais, Instituto Socioambiental e dos Recursos Hídricos, Universidade Federal Rural da Amazônia-UFRA, Avenida Presidente Tancredo Neves 2501, Belém CEP 66077-830, PA, Brazil; glauber.palheta@ufra.edu.br (G.D.A.P.); nuno.melo@ufra.edu.br (N.F.A.C.d.M.)

[3] Laboratório de Imunologia e Patologia de Organismos Aquáticos, Instituto de Oceanografia, Campus Carreiros, Universidade Federal do Rio Grande-FURG, Av. Itália, km 8, Rio Grande CEP 96201-900, RS, Brazil; dcluis@yahoo.com

[*] Correspondence: luzrk@vet.ufmg.br; Tel.: +55-31-3409-2218

**Abstract:** The objective was to evaluate the hematological and biochemical blood parameters and performance of *Colossoma macropomum* submitted to BFT maturation and under different feeding regimes. BFT maturation was carried out for 60 days (Phase 1). Feeding on six or seven days a week and feeding rates of 4% or 6% of biomass were tested (Phase 2). The water quality parameters were monitored throughout the experimental period. At the end of Phases 1 and 2, blood samples and zootechnical performance were evaluated. In Phase 1, total ammonia was higher on the 17th day (1.25 mg TAN $L^{-1}$) and stabilized from the 21st day onwards. Nitrite reached a peak (9.67 mg $L^{-1}$) on the 26th day. There was an increase in nitrate between the 25th and 60th day (1.79 ± 0.01 vs. 5.45 ± 0.01 mg N-$NO_3^-$ $L^{-1}$, respectively). FCR (1.90 ± 0.21), weight gain (9.81 ± 1.08 g), and SGR (1.26 ± 0.12%) were highest at 30 days of phase 1. The glucose level (118.23 ± 26.30 mg $dL^{-1}$) was highest on the 30th day. The plasmatic protein (5.36 ± 0.30 g $dL^{-1}$) and alanine aminotransferase (ALT) levels (27.58 ± 6.58 UI $mL^{-1}$) were highest after 60 days. The hemoglobin level (5.77 ± 0.74 g $dL^{-1}$) was lowest after 30 days. In Phase 2, the triglycerides, ALT, and hematocrit levels were different at the end of the experiment under all feeding regimes. Histological analysis of gills showed a normal condition for fish under BFT. It was possible to apply a feeding regime of six days a week and 4% biomass for juveniles, with 43 g on average.

**Keywords:** tambaqui; intensive production; BFT; feeding schemes

## 1. Introduction

Biofloc technology (BFT) has been studied for many fish and shrimp species of commercial interest [1–7]. Classified as an intensive system with minimal water renewal, BFT depends on the generation of bioflocs, which are aggregates of microorganisms [8]. The primary function of the microorganisms in the system is the production of microbial biomass to recycle nutrients, especially nitrogen compounds, by microbial looping through heterotrophic bacteria [8,9] or chemoautotrophic microorganisms [8,10].

Nitrogenous substances such as ammonia, nitrite, and nitrate exist naturally in aquatic environments but generally present increased levels during cultivation [11–16]. Deamination of metabolic proteins produces ammonia, which represents 70–95% of the total of

nitrogen excreted [17], thus constituting the main residue. Ammonia is also produced by deamination of microbial proteins in protein-rich organic matter, such as undigested food, feces, and dead animals [18]. In water, nitrifying bacteria can oxidize ammonia to nitrite and then nitrate, which can be reincorporated in microbial or vegetable protein [19]. Although at a different rate, three nitrogen compounds can negatively impact the metabolism and growth of organisms in cultivation [1,20–23]. In closed systems, until the establishment of the microbial biomass, total ammonia (TAN) and nitrite are expected to peak and can reach critical levels for aquatic organisms [8]. The increased concentration of nitrogen compounds brings losses to animal welfare and, consequently, to the growth performance of captive organisms [24–26].

*Colossoma macropomum*, known as tambaqui or black pacu, is native to the Amazon and represents the most commercially produced native species in South America, although it is also produced in Asia (China, Indonesia, Malaysia, Myanmar, and Vietnam) [27]. Interest in this species has been growing and includes a search for more efficient systems and feeding management. To improve the growth performance of fish, other factors beyond the production system, such as feeding management, need to be considered. Some such management involving feed restriction and rate have been tested in different systems and with different fish species to evaluate compensatory growth without harming welfare [28–31]. In nature, the fish species *C. macropomum*, *Piaractus brachypomus*, and *Piaractus mesopotamicus* go through long periods of feed restriction, and protocols of feed restriction in captivity have demonstrated improved growth performance [32–35] and reduced production cost with lower labor and feed cost [36–38]. Although the growth performance of juvenile *C. macropomum* in BFT systems has been described [39–41], the physiological condition and growth performance of *C. macropomum* during BFT maturation and when submitted to different feeding regimes with restriction in BFT have yet to be evaluated.

Therefore, this study aimed to investigate, through hematological analysis and growth performance, the physiological response of juvenile *Colossoma macropomum* during BFT maturation and when submitted to different feed regimes under BFT.

## 2. Materials and Methods

### 2.1. Experimental Conditions

The experiment was carried out in the Laboratório de Aquacultura at the Federal University of Minas Gerais (UFMG) following a protocol approved by the Committee for Ethics in Animals Use (CEUA-244/2020) and was divided into two experimental phases.

#### 2.1.1. Phase 1—Growth Performance under Biofloc Systems

Phase 1 lasted 60 days and used 192 juvenile *C. macropomum* (21.25 $\pm$ 0.40 g, 11.19 $\pm$ 0.09 cm) distributed among 16 tanks (n = 12 fish tank$^{-1}$) with volumes of 80 L each (3.22 kgm$^{-3}$). The experimental tanks were arranged in a "macrocosm–microcosm" model [42], being interconnected in a 1000 L matrix recirculating macrocosm to keep the biofloc homogenized [38]. Submersible heater thermostats (at 28 °C) were used to control the water temperature. In addition, a submersible pump was installed in the matrix of the macrocosm to maintain the water circulation with water returned by gravity, maintaining a flow in each tank of 3.18 L min$^{-1}$. An artificial substrate for fixing nitrifying bacteria was added to all tanks (20 $\times$ 35 cm Bedean$^{\circledR}$).

All tanks in the system were filled with clear water, and the photoperiod was maintained at 14 h L:10 h D. Unrefined sugar cane [43] was used as carbon source for BFT maturation, with the C/N ratio being maintained at 15:1 in the beginning and 6:1 thereafter [44]. Carbon (sugar) was added whenever $\geq$1 mg L$^{-1}$ of total ammonia nitrogen was measured in the water [8,10]. The juveniles were fed manually two times a day (8 h and 16 h) with commercial feed extruded (Laguna Peixes Brasileiros-Socil, Brazil) with 4 mm of pellet diameter (crude protein—320 g kg$^{-1}$; ethereal extract min.—50 (g kg$^{-1}$); crude fiber—90 g kg$^{-1}$; mineral matter—140 g kg$^{-1}$; calcium min.—15 g kg$^{-1}$; calcium

max.—30 g kg$^{-1}$; phosphorus min.—6000 g kg$^{-1}$; humidity—1500/1000 g kg$^{-1}$) at 3% of biomass [45], with feed adjustment at every biometric. Uneaten feed was collected 30 min after every feeding, dried in an oven (Nova Étic/Ethink) at 55 °C, and weighed to calculate consumption. Biometrics were performed, and survival was evaluated, at the end of Phase 1.

2.1.2. Phase 2—Effect of Different Feeding Regimes on Growth and Physiology Performance of *C. macropomum* Juveniles Reared in BFT

Phase 2 lasted 60 days and used 96 *C. macropomum* juveniles (43.34 ± 0.35 g, 14.49 ± 0.15 cm) from Phase 1 distributed in 16 tanks (80 L volume) containing 100% inoculum from Phase 1, with density adjusted to n = 6 juveniles tank$^{-1}$ (3.25 kg m$^{-3}$). The BFT system was disposed in the same "macrocosm–microcosm" model as in Phase 1, and the fish were submitted to the following different regimes of weekly feeding frequency and feeding rate in a 2 × 2 factorial, with four repetitions:$T_{6 \times 4}$: fed six days a week, 4% of biomass; $T_{6 \times 6}$: fed six days a week, 6% of biomass; $T_{7 \times 4}$: fed seven days a week, 4% of biomass; $T_{7 \times 6}$: fed seven days a week, 6% biomass.

Juveniles were fed twice a day (8 h and 16 h) with commercial extruded diet (32% crude protein, 4 mm diameter), with feed adjustment after every biometric as determined by biomass. Uneaten feed was collected 30 min after every feeding, dried in an oven at 55 °C, and weighed to calculate consumption. The juveniles were weighed and counted at the end of Phase 2 to calculate survival. The photoperiod was maintained at 14 h L:10 h D, and artificial substrate (20 × 35 cm Bedean$^{®}$) was used in all treatment tanks.

*2.2. Water Quality Analysis*

In Phase 1, total ammonia, nitrite, pH, settleable solids, temperature, and dissolved oxygen were monitored daily, while nitrate was measured on days 25 and 60. In Phase 2, pH, settleable solids, temperature, and dissolved oxygen were monitored daily; total ammonia and nitrite, three times a week; and nitrate, at the beginning and at days 30 and 60. Total ammonia (TAN) [46], nitrite (N-NO$_2$$^-$) [47], and nitrate (N-NO$_3$$^-$) [48]; pH; temperature; and dissolved oxygen (DO) (model 550A YSI multiparameter—Ohio, USA) were monitored from microcosms in quadruplicate (four water samples), while settleable solids (SS) [8,49] (during 15 min) and alkalinity (ALPHA, 2012) were monitored from the macrocosm in duplicate.

For both phases, clarification was performed when settleable solids exceeded 10 mL L$^{-1}$ [50]. Dolomitic limestone was used to adjust alkalinity when it reached ≤100 mg CaCO$_3$$^-$ [51].

Water salinity was maintained at 2 g L$^{-1}$ (checked with refractometer) during the Phase 1 and Phase 2 [52,53].

*2.3. Blood Analysis*

To determine any physiological effects on *C. macropomum* juveniles during biofloc maturation in Phase 1, blood was collected from 12 juveniles at 0 (basal), 30, and 60 days of rearing. To evaluate hematological and biochemical responses of *C. macropomum* juveniles reared in BFT under different feeding regimes in Phase 2, blood was collected from 12 juveniles at the beginning and end of the experimental period. Blood was collected from the caudal vein using heparinized syringes. Part of each blood sample was used to evaluate the hematocrit via capillary tubes [54]. Hemoglobin concentration was determined using 4 µL of blood in 1 mL of commercial colorimetric reagents (Bioclin$^{®}$ Brazil). Plasmatic protein was measured with a manual refractometer (RHC 200-ATC, Huake Instrument Co) after breaking the microhematocrit tube from the hematocrit. Biochemical levels of the glucose, triglycerides, and cholesterol, AST, and ALT were determined by centrifuging blood at 4000 rpm for 10 min for plasma separation and dosage in commercials kits (Bioclin$^{®}$ Minas Gerais, Brazil). All samples were read using a spectrophotometer (Libra S22 Biochrom Cambridge, UK).

### 2.4. Growth Performance

Growth performance was determined for all *C. macropomum* juveniles by measuring weight using a precision scale (0.01 g, AD5002 Marte Minas Gerais, Brazil) and length using an ichthyometer on days 1, 30, and 60 of Phase 1 and days 1 and 60 of Phase 2 to determine:

a.  Weight gain (WG; g) = final weight — initial weight;
b.  Specific growth rate (SGR; % / day) = ((ln final weight — ln initial weight)/days) $\times$ 100;
c.  Production (kg m$^{-3}$) = biomass/tank volume in m$^3$;
d.  Biomass (g) = total number of fish $\times$ final weight;
e.  Feed intake (g) = weight of offered feed — weight of uneaten feed;
f.  Feed conversion rate (FCR) = feed intake/weight gain;
g.  Survival (%) = (number of fish at the end of the experiment/number of initial fish) $\times$ 100.

### 2.5. Indexes

For phase 2, after blood collection, the 12 animals were euthanized (285 mg eugenol L$^{-1}$) [55] and had their livers and adipose tissue removed to calculate their hepatosomatic and mesenteric fat indices as: Hepatosomatic index (HSI) = (liver weight/body weight) $\times$ 100; and Mesenteric fat index (MFI) = (mesenteric weight/body weight) $\times$ 100.

### 2.6. Histology

At the end of Phase 2, samples of gills from six fish of each treatment were fixed in Bouin's fluid for histological processing. All samples were processed using LUPE PT05 automated equipment and were embedded in Paraplast®. Sections were cut at 3 μm with a LUPE MRP03 microtome. Histological sections were stained with hematoxylin–eosin. For classification purposes, light hyperplasia was designated to be 2 to 5 cell layers on the lamellae, moderate hyperplasia was designated to be 5 to 10, and severe hyperplasia was designated to be 10 or more [56].

### 2.7. Statistical Analysis

Data were tested for normality (Shapiro–Wilks) and homoscedasticity (Levene). Prior to analysis, percentage values for both phases were transformed by arc sin square root, while only untransformed data are shown. Growth performance data for Phase 1 were analyzed at days 30 and 60 during biofloc maturation by the Mann–Whitney test for nonparametric data and Student's T-test for parametric data ($\alpha$ = 0.05). Blood data for Phase 1 were analyzed at days 0 (basal), 30, and 60 during biofloc maturation by one-way ANOVA and Tukey´s post hoc test ($\alpha$ = 0.05). Hepatosomatic and mesenteric fat indices for Phase 2 were also submitted to one-way ANOVA and Tukey´s post hoc test ($\alpha$ = 0.05). Growth performance and blood parameters were submitted to factorial (two-way) ANOVA and Tukey´s post hoc test ($\alpha$ = 0.05). Nonparametric data of both Phase 1 and Phase 2 were analyzed by the Kruskal–Wallis test ($\alpha$ = 0.05).

## 3. Results

### 3.1. Phase 1—Growth Performance under Biofloc Systems

Values for temperature, DO, alkalinity, and settleable solids during Phase 1 were 26.9 $\pm$ 0.99 °C, 5.97 $\pm$ 0.32 mg L$^{-1}$, 109.47 $\pm$ 36.43 mg CaCO$_3$ L$^{-1}$, and 3.46 $\pm$ 4.35 mL L$^{-1}$, respectively. Data for TAN, nitrite, and pH are shown Figure 1. The concentration of TAN was highest on day 17 (1.25 mg TAN L$^{-1}$) and stabilized from day 21 onward, with means $\leq$ 0.25 mg L$^{-1}$. Nitrite reached a peak (9.67 mg L$^{-1}$) on day 26 and then reduced, reaching stability on day 43. Nitrate increased significantly ($p < 0.05$) between days 25 and 60 (1.79 $\pm$ 0.01 and 5.45 $\pm$ 0.01 mg N-NO$_3^-$ L$^{-1}$, respectively). During the same period of continuous nitrite increase, pH decreased until day 42, after which it increased. Data for DO are shown in Figure 1B. Dolomitic limestone was added to the BFT on days 33 to 46 to correct alkalinity. Settleable solids were detected starting on day 12 (0.2 mL L$^{-1}$), with a peak (15 mL L$^{-1}$) on day 45, the same period when the system was clarified. Partial

renewal of water (10% of total volume, total of 1300 L) was necessary when nitrite reached $\geq 5$ mg L$^{-1}$.

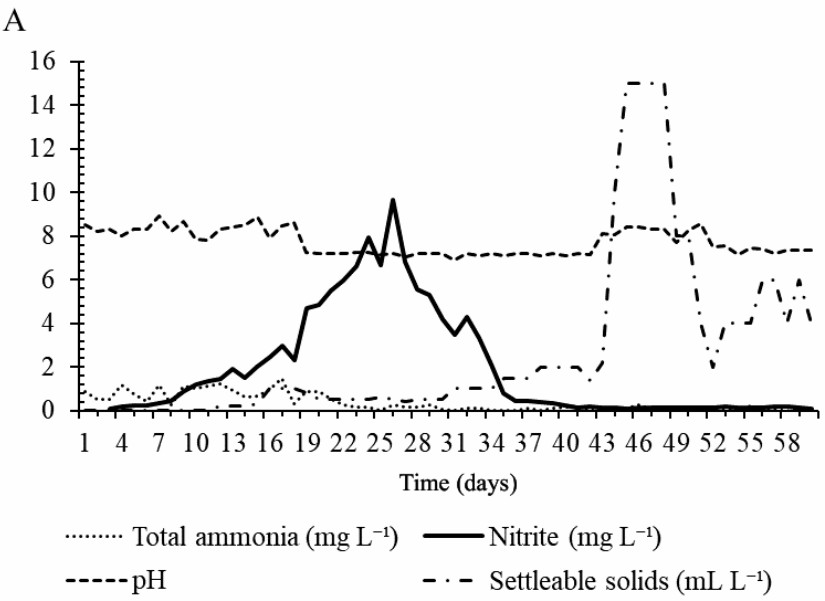

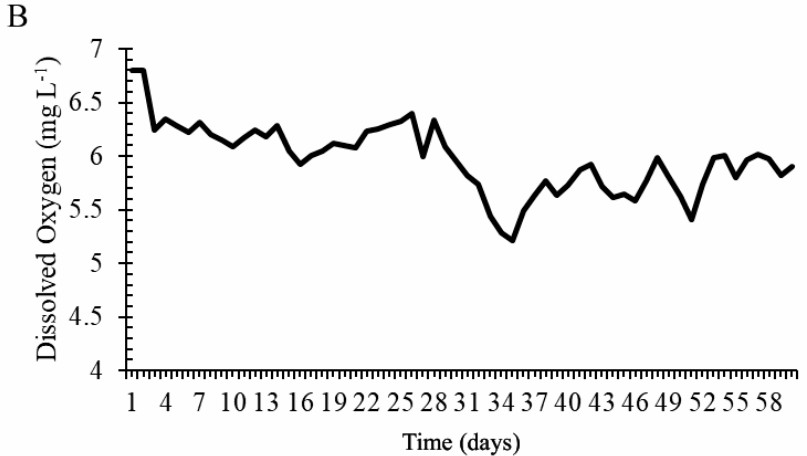

**Figure 1.** (**A**) Curves for total ammonia (mg L$^{-1}$), nitrite (mg L$^{-1}$), settleable solids (mL L$^{-1}$), and pH during biofloc maturation when rearing *Colossoma macropomum* juveniles. (**B**) Curve for dissolved oxygen (mg/L) during biofloc maturation when rearing *Colossoma macropomum* juveniles.

During biofloc maturation, the *C. macropomum* juveniles grew (in both weight and total length) and achieved their highest biomass, and consequently highest production (kg m$^{-3}$), at the end of the experimental period ($p < 0.05$) (Table 1); however, the best performance for FCR, WG, and SGR ($p < 0.05$) was at day 30. Survival did not differ throughout the experimental period ($p > 0.05$).

Changes occurred in the blood parameters of the *C. macropomum* during biofloc maturation (Figure 2). Glucose concentration (Figure 2A) was highest on day 30 ($p < 0.05$). Plasmatic protein (Figure 2E) and ALT (Figure 2G) were highest at the end of the experimental period ($p < 0.05$). Hemoglobin (Figure 2C) differed significantly ($p < 0.05$) among basal, day 30, and day 60, being lowest on day 30. Hematocrit decreased ($p < 0.05$) on day 60 (Figure 2F). Metabolic triglycerides (Figure 2B), cholesterol (Figure 2D), and AST (Figure 2H) did not change significantly during biofloc maturation ($p > 0.05$).

**Table 1.** Values (mean ± standard deviation) for weight, total length, biomass, feed conversion rate, weight gain, feed intake, specific growth rate, and survival of juvenile *Colossoma macropomum* at different times (days 30 and 60) during biofloc maturation.

| Performance | Day 30 | Day 60 |
|---|---|---|
| Weight (W) (g) [1] | 31.06 ± 1.03 [b] | 37.71 ± 2.88 [a] |
| Total length (L) (cm) [2] | 13.19 ± 0.29 [b] | 13.95 ± 0.32 [a] |
| Biomass (g) [1] | 372.68 ± 12.40 [b] | 452.49 ± 34.61 [a] |
| Production (kg m$^{-3}$) [1] | 4.66 ± 0.15 [b] | 5.42 ± 0.58 [a] |
| Feed conversion rate (FCR) [1] | 1.90 ± 0.22 [b] | 3.14 ± 1.27 [a] |
| Freed intake (FI) (g) [1] | 18.46 ± 0.34 [b] | 30.33 ± 2.94 [a] |
| Weight gain (WG) (g) [1] | 9.81 ± 1.11 [a] | 6.67 ± 2.69 [b] |
| Specific growth rate (SGR) (%) [1] | 1.26 ± 0.13 [a] | 0.65 ± 0.24 [b] |
| Survival (%) [1] | 100.00 [a] | 95.83 ± 6.80 [a] |

Different superscript letters in the same row represent significant differences ($p < 0,05$). [1] Mann–Whitney test; [2] Student's *t* test.

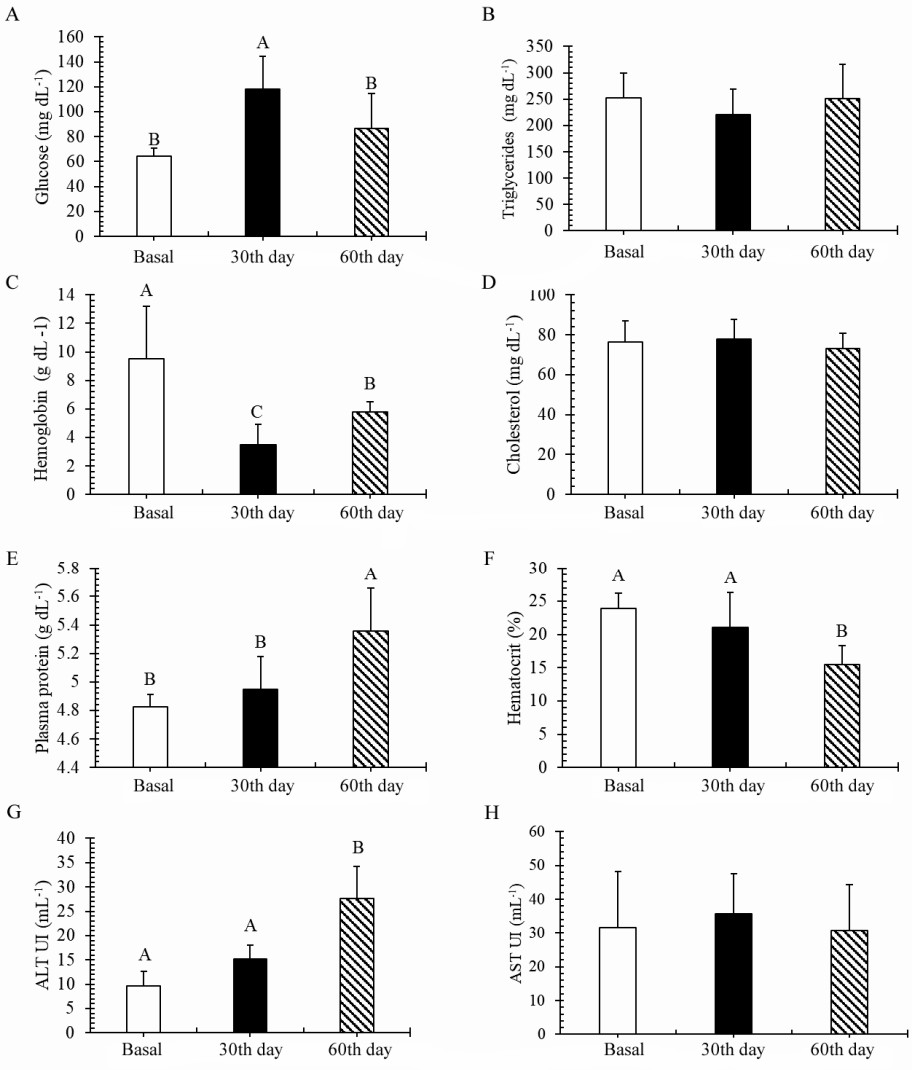

**Figure 2.** Hematological, biochemical, and enzymatic blood parameters of the *Colossoma macropomum* juveniles (n = 12 pre collection) on days 0 (basal), 30, and 60 of rearing under biofloc maturation. Different letters on the bars indicate significant differences over time in glucose (**A**), cholesterol (**D**), plasmatic protein (**E**), hemoglobin (**C**), and ALT (**G**) according to Kruskal–Wallis test ($p < 0.05$). Different letters on the bars indicate significant differences over time in AST (**H**), triglycerides (**B**), and hematocrit (**F**), according to Tukey´s test ($p < 0.05$).

### 3.2. Phase 2—Effect of Different Feeding Regimes on Growth and Physiology Performance of C. macropomum Juveniles Reared in BFT

No differences ($p > 0.05$) were detected for the water quality parameters of temperature ($28.78 \pm 0.09$ °C), pH ($8.43 \pm 0.33$), TAN ($0.12 \pm 0.08$ mg TAN L$^{-1}$), nitrite ($0.18 \pm 0.06$ mg N-NO$_2^-$ L$^{-1}$), DO ($4.92 \pm 0.18$ mg OD L$^{-1}$), alkalinity ($153.08 \pm 34.61$ mg CaCO$_3$ L$^{-1}$), or settleable solids ($5.69 \pm 3.99$ mL L$^{-1}$) among different feeding regimes ($p > 0.05$). Nitrate did not differ significantly among feeding regime treatments ($p > 0.05$) at the beginning or on day 30 or 60 ($5.45 \pm 0.01$, $19.74 \pm 0.11$, and $44.36 \pm 0.11$ mg N-NO$_3^-$ L$^{-1}$, respectively).

The parameters of weight, length, biomass, production, FCR, FI, WG, and SGR did not differ according to weekly feeding frequency or feeding rate after 30 and 60 days, and there was no interaction among them ($p > 0.05$) (Table 2). Survival was 100% for all treatments on day 30 and was $91.65 \pm 9.64$ ($T_{6 \times 4}$), $95.82 \pm 8.35$ ($T_{6 \times 6}$), $91.65 \pm 9.64$ ($T_{7 \times 4}$), and $87.50 \pm 15.94$ % ($T_{7 \times 6}$) on day 60, with no significant differences ($p > 0.05$).

**Table 2.** Growth performance (mean $\pm$ SD) for weight (W, g), total length (L, cm), biomass (g), production (kg m$^{-3}$), feed conversion rate (FCR), feed intake (FI, g), weight gain (WG, g), and specific growth rate (SGR, %) for *Colossoma macropomum* juveniles grown under BFT with different weekly feeding frequencies and feeding rates.

| Day | Feeding Frequency | Feeding Rate | Treatment | W (g) | L (cm) | Biomass (g) | Production (kg m$^{-3}$) | FCR | FI (g) | WG (g) | SGR (%) |
|---|---|---|---|---|---|---|---|---|---|---|---|
| 30 | 6 | 4 | $T_{6 \times 4}$ | 54.65 ± 3.56 | 15.62 ± 0.81 | 327.92 ± 21.38 | 4.09 ± 0.27 | 2.24 ± 0.28 | 24.83 ± 5.46 | 11.37 ± 3.43 | 0.77 ± 0.21 |
|  |  | 6 | $T_{6 \times 6}$ | 58.94 ± 5.21 | 16.07 ± 0.52 | 353.65 ± 31.30 | 4.42 ± 0.39 | 2.03 ± 0.37 | 29.95 ± 5.76 | 15.41 ± 5.36 | 1.00 ± 0.31 |
|  | 7 | 4 | $T_{7 \times 4}$ | 65.06 ± 8.14 | 16.38 ± 0.98 | 390.40 ± 48.87 | 4.88 ± 0.61 | 1.64 ± 0.22 | 34.37 ± 10.35 | 21.70 ± 7.90 | 1.33 ± 0.42 |
|  |  | 6 | $T_{7 \times 6}$ | 57.97 ± 10.58 | 16.05 ± 1.10 | 347.87 ± 63.48 | 4.35 ± 0.79 | 3.14 ± 2.07 | 34.71 ± 8.05 | 14.79 ± 10.63 | 0.94 ± 0.61 |
|  | Feeding frequency *(P)* |  |  | 0.225 | 0.422 | 0.225 | 0.225 | 0.481 | 0.087 | 0.212 | 0.376 |
|  | Feeding rate *(P)* |  |  | 0.712 | 0.894 | 0.711 | 0.711 | 0.167 | 0.490 | 0.704 | 0.674 |
|  | Interaction *(P)* |  |  | 0.149 | 0.395 | 0.149 | 0.149 | 0.089 | 0.544 | 0.162 | 0.156 |
| 60 | 6 | 4 | $T_{6 \times 4}$ | 69.90 ± 8.95 | 17.18 ± 0.35 | 419.42 ± 53.74 | 5.24 ± 0.67 | 2.77 ± 0.61 | 45.09 ± 6.48 | 15.25 ± 6.05 | 0.80 ± 0.34 |
|  |  | 6 | $T_{6 \times 6}$ | 72.69 ± 14.43 | 17.14 ± 1.20 | 436.17 ± 85.56 | 5.45 ± 1.08 | 3.56 ± 2.08 | 50.94 ± 3.43 | 18.64 ± 2.39 | 0.89 ± 0.10 |
|  | 7 | 4 | $T_{7 \times 4}$ | 82.24 ± 5.20 | 18.29 ± 0.86 | 493.46 ± 31.21 | 6.17 ± 0.39 | 2.13 ± 0.14 | 56.07 ± 11.32 | 17.17 ± 4.94 | 0.79 ± 0.36 |
|  |  | 6 | $T_{7 \times 6}$ | 78.21 ± 14.84 | 18.22 ± 1.10 | 469.27 ± 89.07 | 5.86 ± 1.11 | 2.39 ± 0.51 | 62.63 ± 16.57 | 20.23 ± 6.33 | 0.99 ± 0.35 |
|  | Feeding frequency *(P)* |  |  | 0.301 | 0.076 | 0.301 | 0.981 | 0.315 | 0.09 | 0.292 | 0.855 |
|  | Feeding rate *(P)* |  |  | 0.587 | 0.612 | 0.587 | 0.563 | 0.524 | 0.074 | 0.581 | 0.380 |
|  | Interaction *(P)* |  |  | 0.202 | 0.513 | 0.202 | 0.205 | 0.091 | 0.259 | 0.208 | 0.792 |

Two-way ANOVA, Tukey's test ($p < 0.05$).

Plasmatic analysis found glucose (Figure 3A), hemoglobin (Figure 3D), cholesterol (Figure 3C), protein (Figure 3E), and AST (Figure 3H) not to differ among feeding regimes ($p > 0.05$). Triglycerides (Figure 3B), ALT (Figure 3G), and hematocrit (Figure 3F) were significantly different at the end of the experiment ($p < 0.05$) for all feeding regimes, with final concentrations of triglycerides and ALT being lower ($p < 0.05$), and hematocrit higher ($p < 0.05$), under all feeding regimes. There were no significant differences ($p > 0.05$) among treatments for HSI ($1.48 \pm 0.24$) or MFI ($2.46 \pm 0.70$).

Histological analysis revealed hyperplasia for all treatments, all of them were classified as light to moderate (Figure 4B, $T_{7 \times 4}$). Epithelial lifting (Figure 4C, $T_{6 \times 6}$) and lamellar fusion also were observed. In general, all treatments exhibited normal predominances of primary and secondary lamellae (Figure 4A,D, $T_{6 \times 4}$ and $T_{7 \times 6}$).

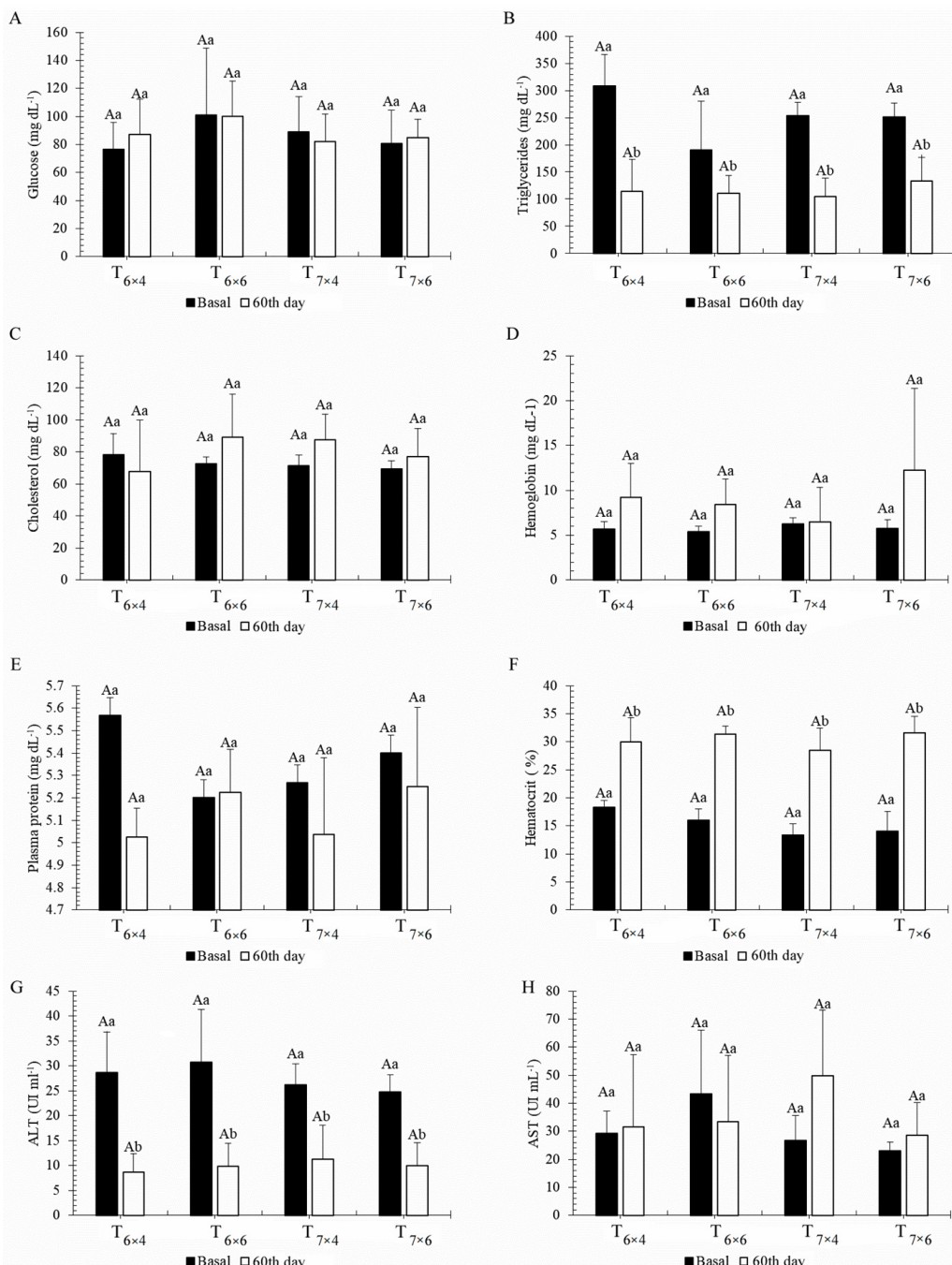

**Figure 3.** Blood parameters (basal and on day 60, Phase I) for *Colossoma macropomum* juveniles under different feeding frequencies and feeding rates: glucose (**A**), triglycerides (**B**), cholesterol (**C**), hemoglobin (**D**), plasmatic protein (**E**), hematocrit (**F**), ALT (**G**), and AST (**H**). Capital letters indicate significant differences between feeding regimes (treatments); lowercase letters indicate significant differences over time (between basal and day 60). Two-way ANOVA, Tukey's test ($p < 0.05$).

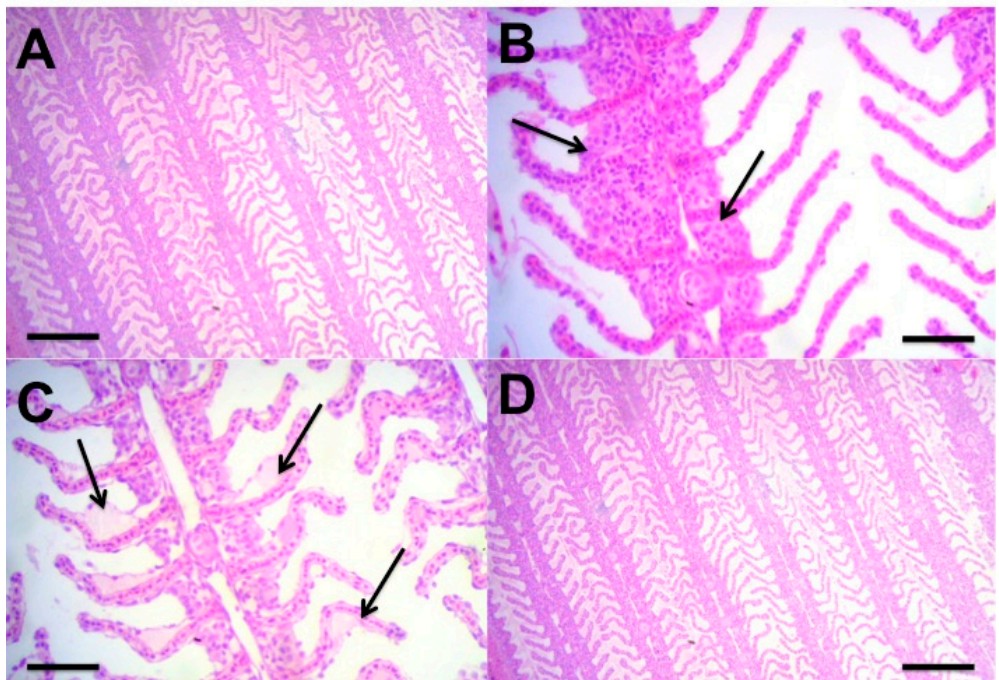

**Figure 4.** Histology of fish gill structure of the *Colossoma macropomum* in BFT at the end of phase 2. (**A**) $T_{6 \times 4}$ and (**D**) $T_{7 \times 6}$, normal primary and secondary lamellae; (**B**) $T_{7 \times 4}$, moderate hyperplasia; and (**C**) $T_{6 \times 6}$, epithelium lifting. (**A**) H–E bar, 200 μ. (**B**) H–E bar, 50 μ. (**C**) H–E bar, 50 μ. (**D**) H–E bar, 200 μ.

## 4. Discussion

In Phase 1, the water quality parameters of temperature, DO, and pH remained within values indicated to the *C. macropomum* [57], and the alkalinity and SS remained within values indicated for fish [8,26].

Analysis of the curves for nitrogen compounds in Phase 1 revealed that they maintained the maturation pattern described in other works with BFT, with a nitrite peak followed by reduction [58–60]. Both TAN and $N\text{-}NO_3^-$ remained at safe levels [61] for fish. Although $N\text{-}NO_2^-$, a nitrogenous compound that causes most of the problems for BFT [62], reached levels considered high for the species [63], in the present study, mortality was not recorded. This could be explained by the short exposure time associated with the use of salt (NaCl). Both $Cl^-$ and $N\text{-}NO_2^-$ have the same inflow route, and increased chloride in water promotes reduction in the inflow of $N\text{-}NO_2^-$, thus decreasing the $N\text{-}NO_2^-$ concentration in blood plasma [64,65]. The $N\text{-}NO_2^-$ in the *C. macropomum* juveniles may not have caused mortality during the period, but effects of peak $N\text{-}NO_2^-$, such as hyperglycemia and changes in protein levels, which signal stressful situations for fish, were seen in blood analyses. In addition, effects of peak $N\text{-}NO_2^-$ were detected in a reduction in hemoglobin. A similar response was described for tambacus (*Piaractus mesopotamicus* × *Colossoma macropomum*) submitted to toxic levels of $N\text{-}NO_2^-$ [66]. This response is due to the oxygen binding site of hemoglobin becoming occupied by $N\text{-}NO_2^-$, transforming hemoglobin into methemoglobin and impairing oxygen transport [67].

In the present study, with a reduction in $N\text{-}NO_2^-$ to safe levels (from the 35th day onwards), it was possible to verify that at the end of the Phase 1 experimental period, the juveniles were recovering in terms of their hemoglobin levels but did not reach their baseline levels, demonstrating a prolonged effect of $N\text{-}NO_2^-$ on *C. macropomum*. At the end of Phase 1, the reduction in hematocrit could also be attributed to the prolonged toxic effect of $N\text{-}NO_2^-$, since in the peak period, the values were similar to baseline values. This same pattern of hemoglobin and hematocrit reduction was seen in *Brycon cephalus*;

the researchers classified it as a picture of functional and hemolytic anemia caused by the toxic effect of $N\text{-}NO_2^-$ [68]. Another prolonged effect attributed to the toxicity of this nitrogenous compound was an increase in ALT. The elevation of this enzyme in plasma is related to liver damage, and the same pattern has already been verified for *Labeo rohita* submitted to $N\text{-}NO_2^-$ [69].

Regarding the growth parameters, although the animals showed increases in weight, total length, and consequently biomass and productivity at the end of Phase 1, performance (WG, SGR, and FCR) was worse, representing productive losses. This drop in performance has been reported in other studies involving $N\text{-}NO_2^-$ toxicity [69] and in the BFT system for *Brycon orbignianus* [70]. In this phase, it is evident that the negative physiological effects and losses in performance caused by the BFT maturation process and its reflexes in production were prolonged, even being present 3.5 weeks after reducing the level of $N\text{-}NO_2^-$ to a safe concentration for the species. The partial renewal of water during the peak in the maturation phase, the use of biofloc inoculum, and water salinization are some tested techniques that can minimize the toxicity of $N\text{-}NO_2^-$ during the maturation of biofloc [24,70].

In Phase 2, the water quality parameters of DO, temperature, pH and alkalinity for the microcosm–macrocosm system remained the same among treatments and were considered ideal for the growth of tropical species [57,71,72]. With the mature BFT system, nitrogen compounds (TAN, $N\text{-}NO_2^-$, and $N\text{-}NO_3^-$) remained at safe levels [62], with no need for water changes, only replacement for evaporation and losses related to the clarification process (total of 1200 L), which kept SS at safe levels, especially to prevent gill occlusion and increased DO consumption by microorganisms present in the biofloc [8].

With the water quality parameters stabilized in this phase, and within the conditions suitable for the species, any difference in the performance of the animals could be attributed to the varied diet. However, performance did not differ among treatments. Studies with different diets reported that the highest feeding rates had the best performance for *P. mesopotamicus* [32] and *Ictalurus punctatus* [33]. Studying different feeding frequencies and feeding rates for *C. macropomum* in net tank [28] showed that the feeding frequency (two vs. three times a day) did not affect the fishes' performance. However, the feeding rate of 10% weight/day showed best results for growth performance. On the other hand, as in the present study, juveniles of *C. macropomum* in RAS subjected to food restriction once a week performed the same as those subjected to daily feeding treatment [35]. Other species, such as *Cichla monoculus* under food restriction of one day [34] and *Lophiosiluris alexandri* under food restriction of both one and two days [73], did not show differences in performance. These results were similar to those in the present study with *C. macropomum* in BFT. The same was reported [37] with *C. macropomum* submitted to one or even three days of restriction, indicating recovery in growth with resumed feeding. It is possible that the same may have happened in the animals under restriction of one day ($T_{6 \times 4}$ and $T_{6 \times 6}$) in the present study without causing losses in performance.

Food restriction could have caused the mobilization of energy reserves through viscerosomatic fat or the liver, reducing IGS and IHS, but no differences were observed in these indices, indicating that short periods of restriction with or without reduced feeding rates did not promote alteration or mobilization of the fat or liver of *C. macropomum* in BFT, a fact that indicates the possibility of less supply in the amount of food and restriction of one day ($T_{6 \times 4}$) for juveniles of *C. macropomum* in BFT with an initial average weight of 43 g.

The FCR values reported for *C. macropomum* in the literature on food restriction have varied. Comparatively, the FCR values of 2.06–4.14 described in [37] for *C. macropomum* in a net tank were close to those of the present study, whereas [35] presented lower FCR values of 0.57–0.8 for *C. macropomum* in RAS, demonstrating that it is possible to improve the efficiency of food use in intensive systems and that this point should be better studied in BFT.

The high survival rates in the present study were in agreement with other works with *C. macropomum* in RAS, net tanks, and nursery systems [28,35,74] demonstrating the plasticity that the species has in adapting to different production systems, including BFT, and different restriction regimes.

BFT has been used as a strategy for maintaining water quality, but it is known that the microorganisms that make up the biofloc can be used as food for fish and shrimp [25,75–77]. In the present study, the lower rates of feeding and/or restriction of one day may have also indicated that juveniles of *C. macropomum* took advantage of the biofloc as food because of the characteristics of the species's feeding habits and the absence of feed supply, which would support the similar performance among treatments, although future studies are needed to confirm the use of biofloc by juveniles of *C. macropomum*.

There were no differences in blood parameters among treatments, even when the *C. macropomum* juveniles were subjected to one-day food restriction in Phase 2. At the end of the experimental period, all treatments showed reductions in triglycerides and ALT. Considering the two phases of the work, the fall in triglycerides and ALT in Phase 2 seemed to be related to recovery after coping with the $N\text{-}NO_2^-$ peak in Phase 1 and not to the effect of different diets, since all treatments presented the same pattern of reductions in both triglycerides and ALT. Changes in glucose concentrations have been an important hematological parameter for defining the condition of fish [78]. In the case of food restriction, and depending on the time of restriction, fish may present hypo- or hyperglycemia [35]. In the present study, glucose levels remained stable among treatments at the beginning and end of the Phase 2 experimental period, and the normal gill histology condition suggested the adaptation of the species to the different feed regimes in BFT. Although damage was observed in the gills of *C. macropomum*, the damage was low and common in fish, e.g., carp [5], without prejudice to *C. macropomum* in BFT systems.

In general, it is possible to indicate the least frequent diet and the lowest feeding rate, $T_{6 \times 4}$, for *C. macropomum* in BFT, resulting directly in reductions in feeding per day and economic costs without damage to animals or production.

## 5. Conclusions

In Phase 1, hematological effects related to stress and performance loss were evident for *C. macropomum* when passing through the $N\text{-}NO_2^-$ peak during BFT maturation. This demonstrates the importance of water management with techniques that reduce $N\text{-}NO_2^-$ during this phase, such as more frequent water changes, increased salinity, and/or the use of biofloc inoculum. In Phase 2, the different feeding regimes demonstrated that it was possible to feed the *C. macropomum* juveniles (43 g) six days a week with a feeding rate of 4% of biomass without compromising their performance or hematological condition.

**Author Contributions:** C.L.N.: Conceptualization, methodology, software, validation, formal analysis, investigation, resources, data curation, writing—original draft preparation, writing—review and editing, visualization, supervision, project administration, funding acquisition. L.F.S.S.: Conceptualization, methodology, formal analysis, investigation, resources, data curation, writing—original draft preparation, visualization. F.A.C.d.S.: Conceptualization, methodology, formal analysis, investigation, resources, data curation, writing—original draft preparation, writing—review and editing, visualization, visualization. T.P.B.: Conceptualization, methodology, formal analysis, investigation, resources, data curation, writing—original draft preparation, visualization. G.C.F.: Conceptualization, methodology, software, validation, formal analysis, investigation, resources, data curation, writing—original draft preparation, writing—review and editing, visualization, supervision, project administration, funding acquisition. G.D.A.P.: Conceptualization, methodology, formal analysis, investigation, resources. N.F.A.C.d.M.: Conceptualization, methodology, formal analysis, investigation, resources. L.A.R.: Conceptualization, methodology, formal analysis, investigation, resources, data curation, writing—original draft preparation, visualization. R.K.L.: Conceptualization, methodology, software, validation, formal analysis, investigation, resources, data curation, writing—original draft preparation, writing—review and editing, visualization, supervision, project administration, funding acquisition, funding acquisition. All authors have read and agreed to the published version of the manuscript.

**Funding:** This research was funded by Coordenação de Aperfeiçoamento de Pessoal de Nível Superior (CAPES, Brazil—Procad 88887.200588/2018-00), and the Fundação de Amparo à Pesquisa do Estado de Minas Gerais (FAPEMIG, Brazil). LUZ, R.K. received a research grant from the Conselho Nacional de Desenvolvimento Científico e Tecnológico (CNPq Process 308547/2018-7).

**Institutional Review Board Statement:** The study was conducted in accordance with and approved by the Institutional Ethics Committee of Universidade Federal de Minas Gerais (protocol code on Animal Use is CEUA-244/2020).

**Data Availability Statement:** The data presented in this study are available on request from the corresponding author.

**Acknowledgments:** Conselho Nacional de Desenvolvimento Científico e Tecnológico (CNPq-Brazil), Coordenação de Aperfeiçoamento de Pessoal de Nível Superior (CAPES-Brazil—Procad 88887.200588/ 2018-00), and Fundação de Amparo à Pesquisa do Estado de Minas Gerais (FAPEMIG-Brazil). LUZ, R.K. received a research grant from the Conselho Nacional de Desenvolvimento Científico e Tecnológico (CNPq Process 308547/2018-7).

**Conflicts of Interest:** The authors declare no conflict of interest.

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
