# Peer review of "Zootechnical Performance and Some Physiological Indices of Tambaqui, Colossoma macropomum Juveniles during Biofloc Maturation and in Different Feed Regimes"

_agriculture, doi:10.3390/agriculture12071025_

Round 1

Reviewer 1 Report

This study evaluated just hemoglobin, HCT and some biochemical parameters beside growth performance of pacu. Accordingly, due to the title and aim could be modified to be “Zootechnical performance and some physiological indices of pacu, Colossoma macropomum juveniles during biofloc maturation and in different feed regimes”

However, the establishment of biofloc technology is an ecofriendly and sustainable methods, which could examine and validating for most of culture fish species, the current manuscript need sever revision and improvement, including scientific and style check.

Some comments

The abstract is weak and not express the results, it could be supported with the percent of improvements.

L27: this sentence “C. macropomum juveniles reached the highest biomass at the end of Phase I” is a logic results could you replace it with the SGR which could be used for comparison with other experiments.

 The ingredients and chemical composition of the used diet could be provided in materials and methods or if it commercial diet adds the full details and the chemical composition.

The anesthesia could be considered also during the handling of fish for biometric and blood sampling.

L80 italicized THE Latin name.

L83: the stocking density is  (3.22 kg/m3) not  ( 0.32 kg/m3).

L86: indicated that the tanks were supported with Submersible heater thermostats but the temperature was not provided.

L 124: add space “watersamples”.

L 137: delete this sentence “The blood parameters analyzed in both phases were: glucose, tryglicerids, cholesterol.......” its repeated in line 145.

L 151: rearrange the growth indices and feed utilization “Weight gain, Specific growth rate, Biomass Production, Feed intake, Feed conversion rate.

2.1.1. the title of Phase 1 could be “Growth performance of under biofloc systems, rather than “BFT maturation and growth ...... .

2.1.2. the title of Phase 2 “could be “effect of different feeding regimes on growth and physiology performance of C. macropomum juveniles reared in BFT”

As the second phase of the experiment focused on feeding regime, the histological evaluation could considered the intestine and liver histology.

Figure 4: the histological comparison must use the same scale and magnification.

L273: contain some Latin words “ao final da fase 2”.

The caption of Table 1. Is not correct.

its still early to go through the discussion before improving the general structure of the manuscript "materials and results sections"

Author Response

Dear,

We are very grateful for the dedication and thoughtful comments of the referees, which helped us to improve the quality of the manuscript.

Reviewer 1

  1. Line 21- verb missing

A: Thank you for your comment. Changes were done directly in the manuscript (Line 26).

  1. Line 40 – depends

A: Thank you for your correction. Changes were done directly in the manuscript (Line 46).

  1. Line 54 – ´´reached`` for ´´reach``

A: Thank you for your comment. Changes were done directly in the manuscript (Line 61).

  1. Line 77 - ´´in Laboratorio`` for ´´in the Laboratorio``

A: Thank you for your comment. Changes were done directly in the manuscript (Line 87).

  1. Line 217 - Please insert table legend and the initial values

A: Thank you for your suggestion. Changes were done directly in the manuscript (Line 241-246).

  1. Line 283 - please reformulate

A: Thank you for your comment. Changes were done directly in the manuscript (Line 318).

  1. Line 284 - Please reformulate since it means that you did not record the mortality, but in the light of the following phrase, I understand that there was no mortality

A: Thank you for your comment. Changes were done directly in the manuscript (Line 320).

  1. Line 289 - please reformulate

A: Thank you for your suggestion. Changes were done directly in the manuscript (Line 323).

  1. Line 291 - please reformulate

A: Thank you for your comment. Changes were done directly in the manuscript (Line 323).

  1. Line 308 - ´´grownth`` for ´´the growth parameters``

A: Thank you for your comment and correction. Changes were done directly in the manuscript (Line 344).

  1. Line 330-332 - this phrase is unclear. Please check the grammar

A: Thank you for your comment. Changes were done directly in the manuscript (Line 366-369).

  1. Line 333 - the subject is missing

A: Thank you for your comment. Changes were done directly in the manuscript (Line 372-374).

  1. Line 380-383 - please reformulate

A: Thank you for your comment. Changes were done directly in the manuscript (Line 424-426).

A: Dear Reviewer, we are thankful for the opportunity to publish on periodic, we hope to take into account all the questions, comments, and suggestions.

Reviewer 2 Report

The  paper is interesting. There are some critical issues with the language, therefore I recommend English revision. 

My comments are reported in the file 

Author Response

We are very grateful for the dedication and thoughtful comments of the referees, which helped us to improve the quality of the manuscript.

Reviewer 2 

  1. This study evaluated just hemoglobin, HCT and some biochemical parameters beside growth performance of pacu. Accordingly, due to the title and aim could be modified to be “Zootechnical performance and some physiological indices of pacu, Colossoma macropomum juveniles during biofloc maturation and in different feed regimes”

    A: Thank you for your suggestion. Changes were done directly in the manuscript, we replaced “pacu” by “tambaqui” in the title (Line 2-4).

  1. However, the establishment of biofloc technology is an ecofriendly and sustainable methods, which could examine and validating for most of culture fish species, the current manuscript need sever revision and improvement, including scientific and style check.

    A: Thank you for the opportunity to publish our manuscript, we hope to take into account all the suggestions requested in the review

  1. The abstract is weak and not express the results, it could be supported with the percent of improvements.

A: Thank you for the comment, the abstract was restructured.

  1. L27: this sentence “ macropomum juveniles reached the highest biomass at the end of Phase I” is a logic results could you replace it with the SGR which could be used for comparison with other experiments.

A: Thank you for the suggestion. Changes were done directly in the manuscript.

  1. The ingredients and chemical composition of the used diet could be provided in materials and methods or if it commercial diet adds the full details and the chemical composition.
  2. According with your suggestion, full details of the chemical composition of the commercial diet it was included in the manuscript.

  1. The anesthesia could be considered also during the handling of fish for biometric and blood sampling.

  1. Thank you for observation, but no anesthetic was used to blood collection and biometric samples. And they are in accordance with the protocol approved by the ethics committee.

  1. L80 italicized THE Latin name.

A: Thank you for observation. Change was done in the manuscript.

  1. L83: the stocking density is (3.22 kg/m3) not (≅32 kg/m3).

A: Correct, sorry for the mistake. Change was done in the manuscript.

  1. L86: indicated that the tanks were supported with Submersible heater thermostats but the temperature was not provided.

A: Thank you for observation. We include the temperature value in the manuscript.

  1. L 124: add space “watersamples”.

A: Thank you for the observation, the space has been added.

  1. L 137: delete this sentence “The blood parameters analyzed in both phases were: glucose, tryglicerids, cholesterol.......” its repeated in line 145.

A: Thank you for the suggestion. The sentence has been deleted.

  1. L 151: rearrange the growth indices and feed utilization “Weight gain, Specific growth rate, Biomass Production, Feed intake, Feed conversion rate.

A: Thank you for the suggestion. Changes were done in the manuscript.

  1. 1.1. the title of Phase 1 could be “Growth performance of under biofloc systems, rather than “BFT maturation and growth ...... .

A: Thank you for the suggestion, we have replaced it in the manuscript.

  1. 1.2. the title of Phase 2 “could be “effect of different feeding regimes on growth and physiology performance of C. macropomum juveniles reared in BFT”

A: Thank you for the suggestion, we have replaced it in the manuscript.

  1. As the second phase of the experiment focused on feeding regime, the histological evaluation could considered the intestine and liver histology.

A: Thank you for the suggestion, unfortunately we did not consider collecting theses organs in this experiment, but we will do on next one.

  1. Figure 4: the histological comparison must use the same scale and magnification.

A: Thank you for the comment. After consulting a pathology expert we present the changes in the manuscript.

  1. L273: contain some Latin words “ao final da fase 2”.

A: Sorry for our mistake. Correction it done in the manuscript.

  1. The caption of Table 1. Is not correct.

A: Thank you for the observation. We changed in the manuscript.

  1. its still early to go through the discussion before improving the general structure of the manuscript "materials and results sections"

A: Thank you for the review, we improved general structure.

Round 2

Reviewer 1 Report

the authors addressed all comments carefully,

just modify the title of Phase 1 to: the growth performance of C. macropomum under biofloc systems.